# Haemostatic Nanoparticles-Derived Bioactivity of from *Selaginella tamariscina* Carbonisata

**DOI:** 10.3390/molecules25030446

**Published:** 2020-01-21

**Authors:** Yusheng Zhao, Yue Zhang, Hui Kong, Meiling Zhang, Jinjun Cheng, Juan Luo, Yan Zhao, Huihua Qu

**Affiliations:** 1School of Chinese Materia Medica, Beijing University of Chinese Medicine, Beijing 100029, China; 16688091339@163.com; 2School of Life Sciences, Beijing University of Chinese Medicine, Beijing 100029, China; 3School of Traditional Chinese Medicine, Beijing University of Chinese Medicine, Beijing 100029, China; doris7629@126.com (H.K.); 18811790361@163.com (M.Z.); carlosjjcheng@163.com (J.C.); luojuan1010@163.com (J.L.); zhaoyandr@163.com (Y.Z.); 4Centre of Scientific Experiment, Beijing University of Chinese Medicine, Beijing 100029, China

**Keywords:** *Selaginella pulvinate Carbonisata*, nanoparticles, haemostasis

## Abstract

High-temperature carbonisation is used to prepare many traditional Chinese medicine charcoal drugs, but the bioactive haemostatic substances of these medicines and their mechanisms are still unknown. This study developed and evaluated nanoparticles (NPs) derived from *Selaginella pulvinate Carbonisata* (STC) for the first time. The haemostatic effect of STC-NPs prepared at 300, 350, and 400 °C were investigated in mouse tail amputation and liver scratch experiments. STC-NPs obtained at 400 °C had the strongest haemostatic effect, and were accordingly characterised by ultraviolet–visible spectroscopy, fluorescence spectroscopy, Fourier transform infrared spectroscopy, transmission electron microscopy, high resolution transmission electron microscopy, X-ray diffractometry, and X-ray photoelectron spectroscopy. STC-NPs averaged 1.4–2.8 nm and exhibited a quantum yield of 6.06% at a maximum excitation wavelength of 332 nm and emission at 432 nm. STC-NPs displayed low toxicity against mouse monocyte macrophage RAW 264.7 cells by CCK-8 assay, and STC-NP treatment significantly shortened bleeding time in rat and mouse models. Coagulation assays showed that the haemostatic effects of STC-NPs were related to improving the fibrinogen and platelet contents, as well as decreasing the prothrombin time that resulted from stimulating extrinsic blood coagulation and activating the fibrinogen system. The STC-NPs had remarkable haemostatic effects in the tail amputation and liver scratch models; these effects may be associated with the exogenous coagulation pathway and activation of the brinogen system, according to the evaluation of the mouse coagulation parameters. This novel evaluation supports the material basis of STC use in traditional Chinese medicine, and this article is worthy of study by authors of clinical pharmacy.

## 1. Introduction

Charcoal treatment is a major category of Chinese medicine, with 2000 years of historical application, especially in the treatment of haemorrhagic diseases that show a convergent haemostatic effect [1,2,3]. The haemostatic effect of charcoal has been included in the *Chinese Pharmacopoeia* [4]. Modern pharmacological studies [5] have proven that *Schizonepetae herba carbonisata* can increase the FIB and platelet content in rats to achieve haemostatic purposes. High-temperature carbonisation is a critical process in the preparation of traditional Chinese medicine (TCM) charcoal formulations. However, the bioactive haemostatic substances of charcoal medicine and their mechanisms have not been clearly reported. One study [6] showed that tannins play a major role in the haemostatic effect of carbon drugs. Another study reported that it was a change in flavonoids that was closely related to haemostatic effect enhancement [7]. These inconsistent results cannot reasonably explain the similar haemostatic effects of many different charcoal medicines. We tried to clarify which small molecular compounds were responsible for the effects of *Selaginella pulvinate Carbonisata* (STC), but the results have been uncertain. STC was not detected in the small molecules after carbonization and dialysis. Therefore, this study explains the haemostatic effect of carbonized medicinal materials from the perspective of nanomaterials.

*Selaginella pulvinate* (ST), locally called “Juan Bai”, is widely distributed in China. The Pharmacopoeia of the People’s Republic stipulates that ST is the all dry grass of *Selaginella pulvinate* (Beauv.) Spring or *Selaginella pulvinata* (Hook. Et Grev.) Maxim. ST has been widely used in China as a medicine to improve blood circulation. ST was first recorded as a medicinal plant in *ShengNong’s Herbal Classic*, written more than 2000 years ago. However, its processed product, *Selaginella pulvinate Carbonisata* (STC), possesses opposing haemostatic activity and is clinically used as a TCM haemostatic to treat haematemesis, disintegration, haematochezia, and other bleeding symptoms. Modern medical research [8] has also confirmed that STC is an effective haemostatic after carbonisation. Carbon dots are nanoparticles, usually 1–10 nm in diameter, with proven theranostic activity [9]. In recent years, under the existing concepts of the material basis for traditional Chinese medicine, many scholars [6,10] have attempted to gain an understanding of the haemostatic material foundation of small molecules or metal ions from the perspective of Chinese medicine, but no obvious research progress has been made. Therefore, the haemostatic components of charcoal drugs must be determined. Nanoparticles have been reported [11] to have haemostatic biological activity. The nanotechnology used in this research shows great clinical medicinal value, which provides evidence to support the further investigation of the considerable potential haemostatic drug and effective material basis of TCM.

STC is a type of calcined herb drug that has been used as a haemostatic medicine to promote haemostasis for thousands of years. This study aimed to explore the haemostatic effect of STC and used STC as an example by which to clarify the novel substances generated after the charcoal processing of *Selaginella pulvinate*. By evaluating the physicochemical properties, such as morphological size, elemental components, and functional groups, using a variety of analytical techniques, this study elucidates the differences between ST and STC, in particular, the STC nanoparticles (STC-NPs) created and evaluated herein. The effect of STC-NPs on key parameters involved in haemorheological and coagulation systems were investigated using rat and mouse models.

## 2. Results and Discussion

### 2.1. Preparation Temperature Optimisation

The preparation temperature conditions of STC-NPs were optimised using three different temperatures (i.e., 300 °C, 350 °C, and 400 °C). As shown in Figure 1a, compared to the NS group, the positive group and the STC-NPs groups prepared at different temperatures had haemostatic effects in the tail amputation model. Furthermore, in the liver scratch model, the positive group and the STC-NP groups significantly decreased the liver bleeding time (Figure 1b). The bleeding time of mice intraperitoneally injected with STC-NPs prepared at different temperatures ranged from short to long (300 °C < 350 °C < 400 °C). STC-NPs prepared at 400 °C had the strongest haemostatic effect (significantly shortened bleeding time) and were selected for use in the following experiments.

### 2.2. High-Performance Liquid Chromatography Data Analysis

The chromatogram obtained from the methanol extract of ST contained several compounds, such as amentoflavone (see Figure 2a). However, no significant peaks were observed in the aqueous STC solution (see Figure 2b), indicating that no active small molecule compounds of the ST were detected in the obtained STC-NPs solution [12]. The results of this experiment show that the samples were highly purified by charcoal processing and dialysis.

### 2.3. Characterisation of STC-NPs

As shown in Figure 3a, the TEM microscope image of STC-NPs indicated a uniformly distributed particle size from 1.4 nm to 2.8 nm that was mainly concentrated at approximately 2.0–2.2 nm; the distribution conformed to normal distribution characteristics. The HRTEM results showed that the STC-NP particle was a spherical structure, with well-resolved lattice fringes and a lattice spacing of 0.381 nm (see Figure 3b). This spacing corresponds to that in previously published literature (0.390 nm) [13]. Figure 3c shows that the XRD spectrum of STC-NPs had distinct diffraction peaks (2θ = 22.765°), indicating that STC-NPs were composed of highly amorphous carbon structures [14].

The NP optical properties were characterised using UV–vis and fluorescence spectroscopy. As is shown in Figure 3d, the NPs had a weak absorption peak at approximately 260 nm, corresponding to the π–π* transition of the conjugated C=C bonds and aromatic sp2 domains [15,16]. The fluorescence spectrum of STC-NPs showed a maximum emission wavelength at approximately 432 nm and excitation wavelength at 332 nm (Figure 3e). The QY of STC-NPs was calculated to be 6.06% using quinine sulphate as a reference. The FT-IR spectrum was used to determine NP surface functional groups (Figure 3f). NPs exhibited a broad characteristic peak at 3441 cm^−1^ corresponding to the -O-H stretching vibration. The characteristic absorption at (cm^−1^) 2921, 2860, and 1400 indicated -CH_3_ (stretching), -CH_2_ (stretching), and the stretching vibration peak of C-N, respectively. In addition, NPs displayed the characteristic bands for C=O and C–O–C vibrations at 1633 cm^−1^ and 1048 cm^−1^, respectively, thereby indicating the presence of sp^2^ [17,18]. It was concluded that the STC-NP surface contained carbonyl, hydroxyl, and amidogen functional groups.

The surface and elemental composition of STC-NPs were further studied by X-ray photoelectron spectroscopy (XPS). As shown in Figure 4a, peaks were evident at 400.41 and 532.95 eV, indicating that the quantum point was composed mainly of O (87.21%), C (11.25%), and a small amount of N (1.54%). The XPS spectrum of O1s (Figure 4b) was fitted with two peaks at 532.6 and 533.4 eV, which were attributed to C–OH and O-C=O, respectively. The C1s spectrum (Figure 4c) was divided into three peaks at 284.8, 286.1, and 288.2 eV, which were assigned to C–OH, C=O, and C=N bonds, respectively. The N1s spectrum (Figure 4d) was resolved into two peaks at 399.8 and 401.2 eV. These peaks corresponded to C–N–C and (C)3–N bonds, respectively [19,20]. The experimental results were consistent with the FT-IR characterisation [21,22]. Our FT-IR and XPS analysis showed that STC-NPs contain hydroxyl groups, and that the presence of carbonyl groups can be attributed to the multiphoton activity process from various oxygen-containing functional groups [23].

### 2.4. Cytotoxicity Detection

NP toxicity has always been an essential problem in biological application. The safety of STC-NPs was evaluated by detecting the cell viability of RAW 264.7 cells in a CCK-8 assay. As shown in Figure 5, the STC-NPs were not toxic to cells at concentrations up to 1.25 mg/mL. A concentration of 10 mg/mL decreased RAW 264.7 cell viability. The experimental results showed that the STC-NPs were safe and biocompatible at concentrations up to 1.25 mg/mL.

### 2.5. Haemostatic Effect of STC-NPs

Compared to the NS group (9.07 ± 0.67 min), the HC group (2.33 ± 0.31 min) and the high-, medium- and low-dose STC-NPs groups (2.75 ± 0.62, 1.82 ± 0.26, 3.59 ± 0.41 min, respectively) significantly (*p* < 0.01) reduced tail bleeding time (Figure 6a). Liver bleeding time was the longest in the NS group (7.93 ± 0.55 min). Figure 6b shows that both the positive group (2.68 ± 0.39 min) and the STC-NP groups (H 2.46 ± 0.29, M 2.08 ± 0.27, L 3.61 ± 0.33 min) significantly (*p* < 0.01) reduced liver bleeding time compared to the normal saline group. These results indicated that the STC-NPs had a significant haemostatic effect.

### 2.6. Effect of STC-NPs on the Coagulation System

The haemostatic mechanism of STC-NP activity evaluated by measuring four coagulation parameters (i.e., activated partial thromboplastin time (APTT), thrombin time (TT), prothrombin time (PT), and fibrinogen content (FIB), and platelet counts (PLT)) in rats showed that APTT and TT values were not significantly different among the four treatment groups (Figure 7a,b). As shown in Figure 7c, compared to the NS group, PT values in HC and the STC-NP groups were significantly decreased (p < 0.05). Fibrinogen (FIB) is an important index by which to study the coagulation process. Figure 7d shows that the high-, medium-, and low-dose STC-NPs groups and HC groups showed significantly (*p* < 0.01) increased FIB values (2.29 g/L, 2.35 g/L, 2.28 g/L, and 2.30 g/L, respectively) compared to that of the NS group (2.04 g/L). The number of platelets (PLT) closely reflected the coagulation situation, and STC-NPs had a significant effect on PLT in rats. Compared to the NS group (1053 × 10^9^/L), the high-, medium-, and low-dose STC-NPs groups and the HC group (1151 × 10^9^/L, 1142 × 10^9^/L, 1121 × 10^9^/L, and 1136 × 10^9^/L, respectively) significantly (*p* < 0.05) increased PLT in rats (Figure 7e).

Nanoparticles have many unique properties such as high aqueous solubility, photoluminescence, straightforward functionalisation, and biocompatibility [24,25]. A range of applications has been developed for emerging nanomaterials including drug delivery [26], biological imaging [27], and optical catalysts [28]. NPs have been reported in other TCMs carbonised at high temperature including *Pollen Typhae Carbonisata*, *Cirsium Setosum Carbonisata*, *Schizonepetae Herba Carbonisata*, and *Junci Medulla Carbonisata* [5,29,30,31]. The STC-NPs possessed components similar to those of other widely-studied nanoparticles [32,33]. In the TEM images, the particle size and morphology of STC-NPs are similar to those previously reported of carbon dots [34]. It can be clearly seen that the atomic lattice fringes of the STC-NPs are consistent with that of graphene quantum dots [35]. According to the UV-vis data, the long absorption tail, i.e., up to 600 nm, is due to the low energy transitions within the surface states, created by the surface functional groups of the STC-NPs [36]. This may be the key to affecting their chemical properties. In addition, this study used HPLC to compare the different chemical compositions of ST and STC-NPs to identify component variation between the two. The main active components of ST contain flavonoids and other small molecular substances [37]. In sharp contrast, STC-NPs did not contain small molecule compounds, which, to some extent, excludes the interference of small molecule compounds on haemostatic activity.

The normal physiological haemostasis and anticoagulation mechanism is a complex physiological and biochemical process mainly composed of three interrelated parts: vasoconstriction and platelet response, anticoagulation systems, and fibrinolysis systems [38]. The coagulation process can be roughly divided into three stages: prothrombin activator formation, thrombin formation, and fibrin formation. In the current study, STC-NPs had no effect on TT and APTT values, and therefore, did not affect the endogenous coagulation pathway in rats. However, the increased PT value indicated that STC-NPs might affect the exogenous coagulation pathway. FIB is an important indicator by which to study the coagulation process and the influence of drugs on coagulation, anticoagulation, and fibrinolytic system function [39]. We speculated that the coagulation-promoting activity of STC-NPs might play a role in haemostasis and coagulation by increasing FIB content in the blood, or by inhibiting the fibrinolytic system. STC-NPs significantly increased PLT in rats, and thereby enhanced the haemostatic effect. The haemostatic effect of STC-NPs and their related action mechanisms provide a novel idea for studying the biological activity of novel nanoparticles. The pharmacological experiment data were the most intuitive parameter to reveal the effectiveness of STC-NPs in stopping bleeding. This experiment, for the first time, proved that STC-NPs had a remarkable haemostatic effect, and it provides a preliminary experimental basis for STC-NPs to be used as a new nanoparticle-based haemostatic drug. Further investigations are needed to elucidate the underlying mechanisms of these effects. The discovery of STC-NPs and the demonstration of their haemostatic effect in this paper provide a new rationale for the material research of charcoal drugs. As an emerging nanostructure material in modern science, its pharmacological activity has been preliminary studied in this experiment. STC-NPs have the potential to become a safe and therapeutic drug for haemorrhagic diseases. This article is worthy of study by authors of clinical pharmacy.

## 3. Materials and Methods

### 3.1. Materials

ST was purchased from Beijing Qiancao Herbal Pieces Co., Ltd. (Beijing, China), and STC was prepared in our laboratory. Haemocoagulase (HC) for injection was purchased from Jinzhou Ahon Pharmaceutical Co., Ltd. (Liaoning, China). Dialysis membranes with a molecular weight of 1000 Da were purchased from Beijing Ruida Henghui Technology Development Co., Ltd. (Beijing, China). The cell counting kit (CCK)-8 was purchased from Dojindo Molecular Technologies, Inc. (Kumamoto, Japan). Mouse monocyte macrophage RAW 264.7 cells were purchased with Peking Union Cell Bank (Beijing, China). Pentobarbital sodium and other analytical grade chemical reagents were obtained from Sinopharm Chemical Reagents Beijing (Beijing, China). All experiments were performed using deionised water (DW). Male Kunming mice (32.0 ± 1.0 g) and Sprague Dawley rats (200.0 ± 10.0 g) were purchased from Beijing Jinmuyang Liability Co., Ltd. (Beijing, China) and kept in a well-ventilated room at 24.0 ± 1.0 °C with 55–65% relative humidity and a 12 h light: dark cycle. The animals were given water and food ad libitum. The animal experimental design and protocols used in this study were approved by the Ethics Review Committee for Animal Experimentation at the Beijing University of Chinese Medicine. All the experimental procedures were performed in accordance with the Regulations for the Administration of Affairs Concerning Experimental Animals approved by the State Council of People’s Republic of China.

### 3.2. Preparation of STC-NPs

First, 50 g ST was placed in a crucible and covered with aluminium foil before replacing the lid to form a seal, then calcined using a muffle furnace (TL0612, Beijing Zhong KeAobo Technology Co., Ltd., China) at 300, 350 and 400 °C, respectively, for 1 h. The obtained STC was crushed in a grinder after cooling to room temperature. Finally, the STC was dissolved in DW and boiled twice (1 h each time), then filtered with filter paper to remove the residue. The solution was purified by dialysis for 7 days (molecular weight cut-off of 1.0 kDa), and the obtained STC-NP solution was stored at 4 °C until use. Figure 8 shows the STC-NP preparation process [5].

### 3.3. Haemostatic Experiment

The haemostatic effect of STC-NPs was evaluated using mouse tail amputation and liver scratch models. The preparation temperature conditions of STC-NPs were optimised by testing three different temperatures (300 °C, 350 °C and 400 °C) and the optimal preparation temperature among them was selected to determine the dose-effect relationship of the drug. In brief, mice were divided into the negative control group (normal saline (NS) intraperitoneal injection), the positive group (0.67 KU/kg (HC) subcutaneous injection) and the high-, medium- and low-dose STC-NPs groups (16.67, 8.33 and 4.17 mg/kg, respectively, by intraperitoneal injection). The mice were anesthetised by an intraperitoneal injection of 4% pentobarbital sodium (50 mg/kg) 1 h after treatment. In the mouse tail amputation model, clean surgical scissors were used to cut the tail 1 cm from the tail tip. The time from when the initial cut was made to when bleeding completely halted was recorded. In the liver scratch model, the mice were cut open along the abdominal midline with surgical scissors to expose the liver, which was then propped up with gauze soaked in saline. The left lateral lobe was punctured to a depth of 2 mm with a 2 mL syringe needle, and the wound was then overlaid with filter paper. The time between the onset of bleeding in the punctured liver and the cessation of bleeding was measured. The bleeding in both models was monitored at 30 s intervals until haemostasis was achieved.

### 3.4. HPLC Analysis

The components of the aqueous STC solution obtained at 400 °C and the methanol ST extract were measured using an Agilent 1260 series high-performance liquid chromatography (HPLC) instrument (Agilent, Waldbronn, Germany). A C_18_ column (250 mm × 4.6 mm, Orochem, IL, USA) packed with 5 mm octadecyl-bonded silica (C_18_) was used for STC-NP separation. All samples were filtered with a 0.22 μm cellulose membrane. The mobile phases A and B were 0.1% formic acid solution and methanol, respectively. As previously reported [40], a modified gradient elution program was performed at a flow rate of 1 mL/min as follows: 0–10 min, 10%–15% methanol; 10–20 min, 15–60% methanol; 20–50 min, 60–80%; 50–60 min, 80–85%. The column temperature was 25 °C and the injected sample quantity was maintained at 10 μL. The detection wavelength was set at 330 nm.

### 3.5. Characterisation of STC-NPs

The morphology, particle size distribution and microstructure of STC-NPs obtained at 400°C were examined by TEM (Tecnai G220; FEI Company, USA) at 200 kV, and the lattice spacing and other internal structures were observed by HRTEM (JEN-1230; Japan Electron Optics Laboratory; Japan) and X-ray diffractometer (XRD). The optical information of STC-NPs was analysed using ultraviolet spectrophotometer (CECIL, Cambridge, UK) and fluorescence spectrophotometer (F-4500, Tokyo, Japan). An infrared spectrophotometer (Thermo, California, USA) was used to analyse the distribution of functional groups on the surface of STC-NPs from 4000–400 cm^−1^. The elemental and surface compositions of NPs were observed by X-ray photoelectron spectroscopy (ESCALAB 250Xi, Thermo Fisher Scientific, USA).

### 3.6. Fluorescence Quantum Yield

The fluorescence Quantum yield (QY) of the STC-NPs was measured using quinine sulphate. % QY was 54 in 0.1 M sulfuric acid (H_2_SO_4_) solution as a standard sample. The slit widths of EX and EM were 10 nm and 10 nm, respectively. In order to minimise the reabsorption effect, A_c_ and A_q_ remained below below 0.05 [41]. NP QY was calculated according to the following formula:Q_c_ = Q_q_(A_c_/A_q_) (I_q_/I_c_) (ƞ_c_^2^/ƞ_q_^2^)(1)
where Q represents fluorescence quantum yield, I is the measured integrated emission intensity, and A and ƞ represent the 332 nm absorption value and the refractive index of the solvent respectively. “c” and “q” represent the NPs and the standard, respectively.

### 3.7. Cell cytotoxicity Assay

The cell cytotoxicity of STC-CDs was analysed using RAW 264.7 cells in a CCK-8 assay. The suspension of well-cultured cells was diluted to 1 × 10^5^ cells/mL and spread on a 96-well plate. PBS buffer was added to the surrounding edges, then the cells were incubated in a CO_2_ incubator (37 ℃, 5% CO_2_) for 24 h. Then, the supernatant was removed from the 96-well plate. Dulbecco’s modified Eagle’s medium (DMEM) was added to the control group, and samples of different concentrations were added to the drug groups with 100 μL per well. The 96-well plate was cleaned three times with PBS then incubated in the CO_2_ incubator for an additional 24 h under the same conditions. CCK-8 reagent (10 μL per well) was added to the cultures in the CO_2_ incubator and incubated for 3 h. Absorbance at 450 nm was detected with a microplate reader (Biotek, Vermont, USA). The obtained data were calculated according to the following formula:Cell Viability (% of control) = (Ae − Ab)/(Ac − Ab) × 100(2)

Ae, Ac and Ab represent the absorbance of the drug administration group, control group and bank group, respectively, at 450 nm.

### 3.8. Coagulation Parameter Measurements

A total of 50 rats were randomly divided into the following five groups (ten per group): negative control group (NS), positive drug group (0.67 KU/kg (HC)), and the high-, medium- and low-dose STC-NP groups (16.67, 8.33, and 4.17 mg/kg, respectively). Rats were anaesthetised with a 4% pentobarbital sodium (50 mg/kg) by intraperitoneal injection. Abdominal aortic blood was injected into prepacked 3.2% sodium citrate (blood : citrate : 1:9, *v*/*v*) centrifuge tubes and allowed to react for at least 30 min. Then, the samples were centrifuged at 750× *g* for 15 min to obtain the supernatant. APTT, TT, PT and FIB were measured using an automatic coagulation analyser [42]. PLT was determined by collecting another 60 μL of EDTA-K2 anticoagulant-mixed whole blood.

### 3.9. Statistical Analysis

All experimental data were analysed using the statistical package for the social sciences (SPSS, version 19.0). Normally-distributed and variance homogeneity data were expressed as the mean ± standard deviation (SD). One-way analysis of variance (ANOVA) was used for pairwise comparisons among multiple experimental groups, and non-normally distributed data were expressed as medians (quartile ranges). *p* < 0.05 and *p* < 0.01 indicated statistically significant differences.

## 4. Conclusions

This study is the first to report that novel NPs derived from ST have haemostatic effects in mouse tail amputation and liver scratch models, and that these effects may be associated with the exogenous coagulation pathway and activation of the brinogen system, according to the evaluation of the mouse coagulation parameters. The low toxicity of STC-NPs indicated that they have good prospects for biological application. The haemostatic effect of STC-NPs suggests a new approach for the research and development of therapeutic drugs for haemorrhagic diseases, as well as a novel means by which to explore the mechanism of effective TCM ingredients.

## Figures and Tables

**Figure 1 molecules-25-00446-f001:**
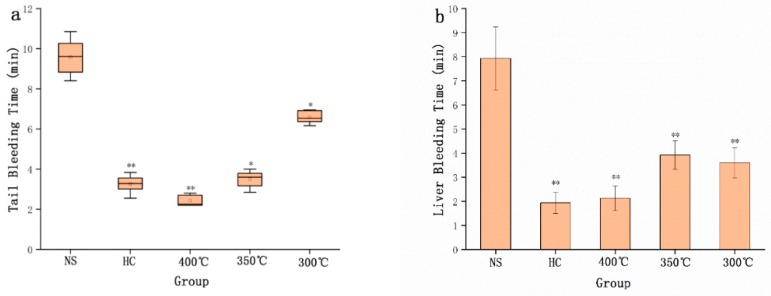
Haemostatic time in mouse tail amputation and liver scratch models: (**a**) Tail amputation (*n* = 8) and (**b**) liver scratch (*n* = 8) models treated with normal saline (NS), haemocoagulase (HC), and STC-NPs produced at 400 °C, 350 °C, and 300 °C. ***p* < 0.01 and **p* < 0.05 compared to the control group (*n* = 8).

**Figure 2 molecules-25-00446-f002:**
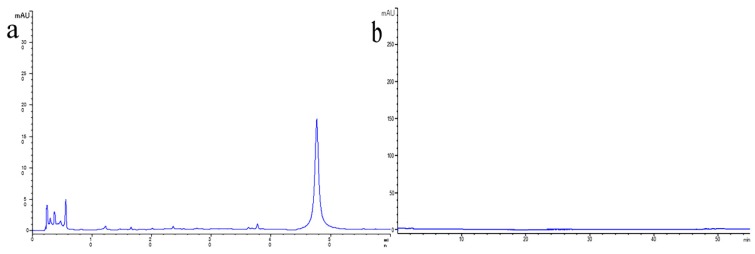
High-performance liquid chromatography (HPLC) fingerprint of (**a**) ST (the all dry grass of *Selaginella pulvinate* (Beauv.) Spring) and (**b**) *Selaginella pulvinate Carbonisata*.

**Figure 3 molecules-25-00446-f003:**
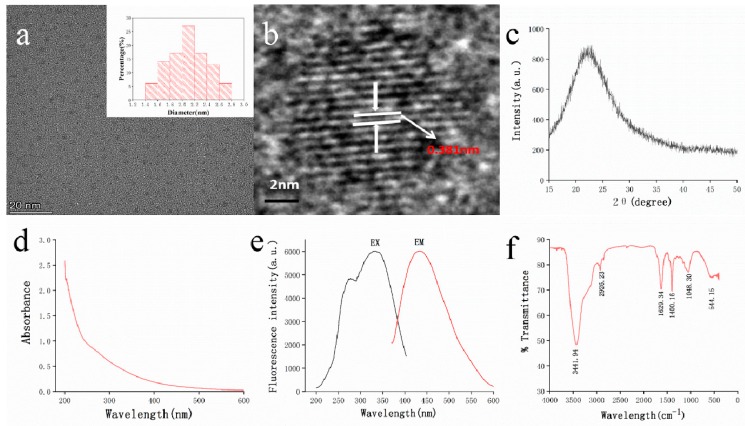
Characterisation of STC-NPs: (**a**) Transmission electron microscopy (TEM) images of STC-NPs displaying ultra-small particles and TEM size distribution of STC-NPs (Upper right corner). (**b**) High-resolution TEM image of STC-NPs. (**c**) X-ray diffraction pattern. (**d**) Ultraviolet–visible spectrum. (**e**) Fluorescence spectrum. (**f**) Fourier transform infra-red spectrum.

**Figure 4 molecules-25-00446-f004:**
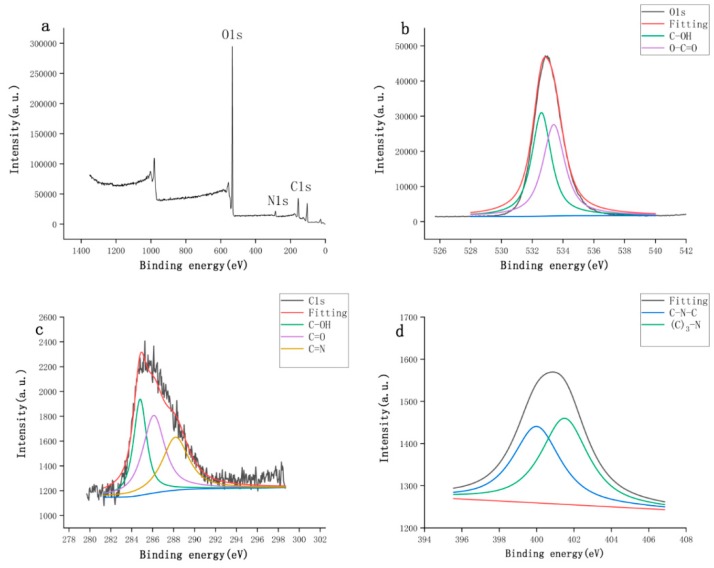
The surface composition and elemental analysis of the prepared STC-NPs by XPS. (**a**) X-ray photoelectron spectroscopy survey of STC-NPs. (**b**) C1s. (**c**) O1s and (**d**) N1s.

**Figure 5 molecules-25-00446-f005:**
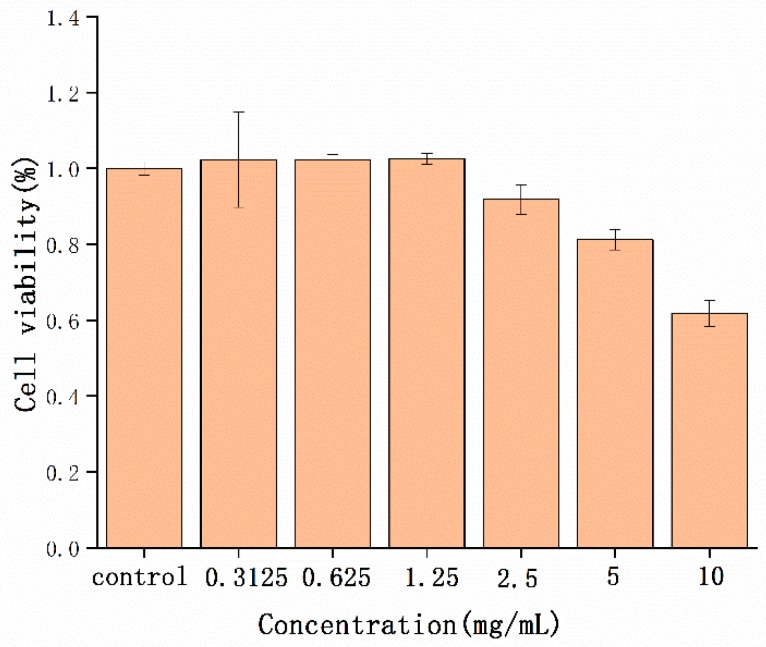
Effect of different concentrations of STC-NPs on the viability of RAW 264.7 cells.

**Figure 6 molecules-25-00446-f006:**
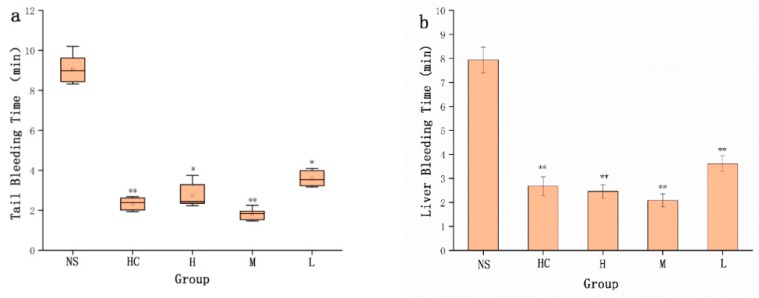
Graphs of haemostatic time in mouse tail amputation and liver scratch models. (**a**) Tail amputation (*n* = 8) and (**b**) liver scratch (*n* = 8) models treated with normal saline (NS), haemocoagulase (HC), and high (H), medium (M), and low (L) doses of STC-NPs (16.67, 8.33, and 4.17 mg/kg, respectively). ***p* < 0.01 and **p* < 0.05 compared to the control group (*n* = 8).

**Figure 7 molecules-25-00446-f007:**
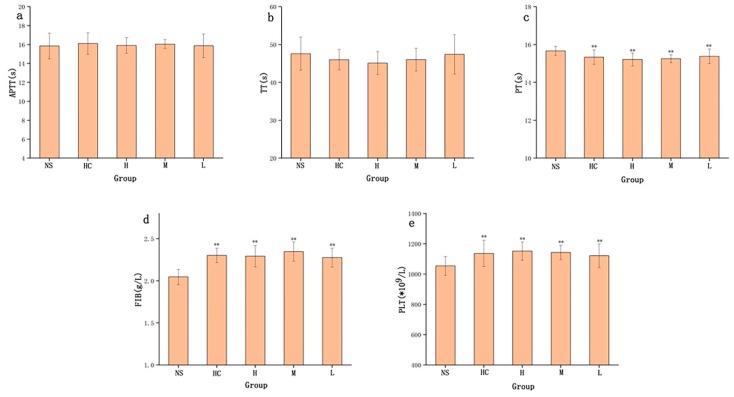
Effects on coagulation parameters. (**a**) Activated partial thromboplastin time (APTT), (**b**) thrombin time (TT), (**c**) prothrombin time (PT), (**d**) fibrinogen (FIB), and (**e**) platelet count (PLT). Analysis of rats treated with normal saline (NS), haemocoagulase (HC), or high (H), medium (M), or low (L) doses of STC-NPs (16.67, 8.33, and 4.17 mg/kg, respectively) ***p* < 0.01 and **p* < 0.05 compared to control group (*n* = 10).

**Figure 8 molecules-25-00446-f008:**
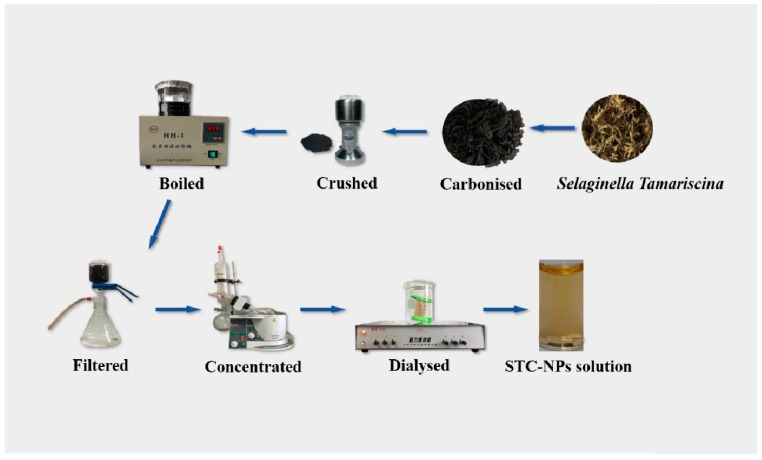
Flow diagram of the STC-NPs preparation process.

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
