# Peer review of "Haemostatic Nanoparticles-Derived Bioactivity of from *Selaginella tamariscina* Carbonisata"

_molecules, 2020, doi:10.3390/molecules25030446_

Round 1
Reviewer 1 Report
The manuscript entitled “Haemostatic bioactivity of nanoparticles derived from Selaginella Tamariscina Carbonisata” is a paper that overviews the utilization of Selaginella Tamariscina a known plant and frequently used in the Chinese medicine, and its hemostatic properties.
The topic is interesting and the paper is well written. The paper is a little hard to follow due to the high number of abbreviations.
Before publication please consider the following comments.
Please use italic every time for Selaginella Tamariscina Carbonisata.
The reviewer recommends to use hemostatic/ hemorrhagic/ hemocoagulase instead of hemostatic/ haemorrhagic/ haemocoagulase, which is more familiar for the readers.
Please leave a space before citation.
Line 22: please specify what type of cells are RAW 264.7 cells
The figures need to stand alone, please specify all the abbreviations in the figure caption (i.e. figure 6 STC-NPs)
Line 41: please use italic for Selaginella tamariscina
Line 45, 78: please use italic for Selaginella Tamariscina
Line 79-83: is there any literature that can sustain the discussion?
Line 161: may be an extra space
Line 164-165: please use italic for the plants names
Line 194-201: please add a specification about the cell line
Line 259: please modify the format
In the bibliographic list, please add a point after the abbreviated name of the journals
Reviewer 2 Report
Abstract: Complete the Abstract with justification of the study, specific conclusions from the study, and a group of readers of the paper who may find it useful.
Keywords: The Keywords should not contain words used in the title of the paper.
Introduction: P. 1, l. 32 – 34 P. Explain the effect of natural product extracts on platelet aggregation activity. Describe the contemporary research on the issues presented in the publication. Smoothly refer to the “carbonizing process of the pollen grains. ... this phenomenon has been confirmed by modern pharmacology research [4]” - explain this in greater detail
35 – 40 Explain the interrelationships between individual groups of biologically active compounds in relation to the topic of the manuscript.l.41 - the name of the species should be italicised (elsewhere in the paper)
2, l. 45 – 48 Which biologically active substances are present in the raw material from Selaginella tamariscina (and other species) to be used in the supportive treatment of haematemesis, disintegration, haematochezia, and other bleeding symptoms. Give the ranges of the content of these phytocompounds, especially those with antioxidative properties, e.g. flavonoids. 49 – 53 Describe exactly reports of contemporary research closely referring to the topic of the study. Indicate what benefits in medicine are provided by the nanotechnology used in relation to the conducted investigations. Describe comparison of the coagulation activity of Selaginella.l.54 – 60. Complete the research hypotheses. Specify the aim of the study accurately, with thorough justification why this type of research was undertaken.
Results and Discussion
L.60 – 65 Fig. 1., Complete the X axis on the graph. Transfer Fig. 1 to the results.
In the description of Fig. 1, there should be Figure 1a, b (according to the guidelines for authors, and change this in all figures in the manuscript analogously). Figure 1a and figure 1b are not cited in the text.
l.92 – 108 Fig. 3d is not cited.
Expand the discussion, which aims to track other results in relation to the results of the present study, by comparison with other literature reports (analogously: P.5. l. 111-119; P. 7. 160 – 182).
P.4. L. 98. In the captions of Figure 3, indicate which of the figures a-f the authors have in mind.
104 in the expression “absorption at 2921 cm-1, 2860 cm-1, and 1400 cm-1”, I suggest using brackets: (cm-1) before the numbers and separate the first two numbers with a semicolon, which will eliminate repetition: “absorption at (cm-1) 2921; 2860, and 1400”P.5. L. 110 Complete the captions of figure 4a-d.
P.6. l. 123- 153 There is no discussion
129 Complete the captions of figure 6a, b. 130 – 140 there is no citation and description of figures 6a, 6b 142-153 there is no citation of figures 7a, 7b, 7c, 7d, 7e, 7f 7. L. 155. Complete the caption of Figure 7a-e. 8. L. 188-191 Indicate the interdisciplinary character of the research and the group of readers that will find it useful.Indicate the subsequent trend that should be followed by future investigations.
Materials and Methods P. 8 -11. provide literature referring to the research method used.
Conclusion: P.11. l. 296 – 300. The conclusions from the study should be formulated clearly and specifically.
References – general notes for use in the text P. 11 - 14: Complete the pages, correct the notation of publication titles and correct the abbreviations of journals, unify the notation of species, etc. The entire text should be read again and corrected in accordance with the guidelines for authors.
Reviewer 3 Report
This study presents a systematic characterization of ST nanoparticles after carbonization and evaluation on their haemostatic properties in experiments on mice and rats. From this study it was revealed that haemostatic properties of STN depends on the carbonization temperature, betterresults being observed for samples treated at higher temperature. He authors analyzed also he cell viability as a function of STN concentration, and using this data they performed further in vivo experiments. In my opinion, the study is well conducted and should be accepted for publication.
